# Revolutionizing the Garment Industry 5.0: Embracing Closed-Loop Design, E-Libraries, and Digital Twins

**Semih Donmezer** [1], **Pinar Demircioglu** [2,3,*], **Ismail Bogrekci** [3], **Gokcen Bas** [1] and **Muhammet Numan Durakbasa** [1]

1 Institute of Production Engineering and Photonic Technologies, Vienna University of Technology (TUWien), 1060 Vienna, Austria; semihdonmezer@gmail.com (S.D.); goekcen.bas@mail.ift.tuwien.ac.at (G.B.); numan.durakbasa@tuwien.ac.at (M.N.D.)

2 Institute of Materials Science, TUM School of Engineering and Design, Technical University of Munich (TUM), 85748 Garching, Germany

3 Mechanical Engineering Department, Engineering Faculty, Aydın Adnan Menderes University, Aydın 09100, Türkiye; ibogrekci@adu.edu.tr

* Correspondence: pinar.demircioglu@tum.de

**Abstract:** This study presents an innovative approach for modernizing the garment industry through the fusion of digital human modeling (DHM), virtual modeling for fit sizing, ergonomic body-size data, and e-library resources. The integration of these elements empowers manufacturers to revolutionize their clothing design and production methods, leading to the delivery of unparalleled fit, comfort, and personalization for a wide range of body shapes and sizes. DHM, known for its precision in representing human bodies virtually and integrating anthropometric data, including ergonomic measurements, enhances the shopping experience by providing valuable insights. Consumers gain access to the knowledge necessary for making tailored clothing choices, thereby enhancing the personalization and satisfaction of their shopping experience. The incorporation of e-library resources takes the garment design approach to a data-driven and customer-centric level. Manufacturers can draw upon a wealth of information regarding body-size diversity, fashion trends, and customer preferences, all sourced from e-libraries. This knowledge supports the creation of a diverse range of sizes and styles, promoting inclusivity and relevance. Beyond improving garment fit, this comprehensive integration streamlines design and production processes by reducing the reliance on physical prototypes. This not only enhances efficiency but also contributes to environmental responsibility, fostering a more sustainable and eco-friendly future for the garment industry and embracing the future of fashion, where technology and data converge to create clothing that authentically fits, resonates with consumers, and aligns with the principles of sustainability. This study developed the mobile application integrating with the information in cloud database in order to present the best-suited garment for the user.

**Keywords:** garment industry; digital twins; DHM (digital human modeling); e-libraries; sustainability; customer centric

## 1. Introduction

Sustainable development [1,2] in the virtual garment industry combines digital innovation with environmentally and socially responsible practices. It aims to reduce the environmental impact of clothing production, promote ethical labor practices, and educate consumers about sustainable choices, all while harnessing the potential of digital technologies to revolutionize the fashion sector. Collaborative efforts among industry stakeholders are essential for achieving a more sustainable and responsible virtual garment industry. In sustainable virtual production, the creation of digital assets or products occurs without the need for physical materials, reducing the consumption of resources and material waste. The garment industry is currently undergoing a transformation through the adoption of advanced technologies such as e-libraries, digital twins, and Industry 5.0. The integration

of e-libraries [3] has introduced a valuable repository of personalized body-size data and historical preferences, empowering garment manufacturers to create tailored and well-fitted garments that cater to the unique needs of individual consumers. The incorporation of robotics presents a promising avenue for embracing data-driven approaches and traditional practices to align with the evolving demands of consumers and the market. This shift towards robotics signifies a new period of technological advancements, reshaping the processes involved in garment production, and delivery to meet the changing the consumer preferences. Table 1 provides an overview of technical features associated with e-libraries, digital twins, and Garment Industry 5.0 (GI5.0) in the garment industry, highlighting how these technologies contribute to automation, customization, and sustainability.

**Table 1.** Technical features comparison: E-libraries, digital twins, and Garment 5.0 in the garment industry. N/A means Not Applicable.

| Parameter | E-Libraries | Digital Twins | Garment 5.0 |
| --- | --- | --- | --- |
| Automation [4] | N/A | Real-time data modeling and analysis | Robotic sewing arms, automated cutting machines |
| Customization [5] | N/A | Tailored virtual replicas for design optimization | Personalized sizing and style recommendations |
| IoT Integration [6] | N/A | Sensors and IoT devices for physical asset tracking | Smart textiles with embedded sensors |
| Artificial intelligence (AI) [7] | AI-powered search and recommendation engines | AI-driven simulations and predictive maintenance | AI-based quality control and production planning |
| Sustainability [8] | Sustainable content curation | Eco-friendly process optimization | Use of sustainable materials and recycling |
| Supply chain transparency [9] | Traceability of information sources | End-to-end visibility in the supply chain | Blockchain-based transparency in the supply chain |
| Digital twins [10] | N/A | Creation of garment digital twins | Virtual prototyping and testing of garments |
| Three-dimensional printing [11] | N/A | Three-dimensional printing for prototyping and production | Three-dimensional-printed custom components and accessories |
| Augmented reality (AR) [12] | N/A | AR-based design visualization and collaboration | AR-powered virtual fitting rooms |
| Blockchain [13] | Secure access and IP protection | Secure data sharing and provenance tracking | Blockchain for supply chain authentication |
| Sustainable materials [14] | Sustainable material databases | Sustainable material recommendations | Use of eco-friendly fabrics and dyes |
| Technical features [15] | Digitalized catalogs, advanced search algorithms | Real-time data synchronization, sensor integration | IoT-enabled production lines, RFID tagging |

The problem addressed in the research study is the need for transformative solutions within the garment industry to address various challenges. These challenges include enhancing sustainability practices, improving production efficiency, reducing waste and environmental impact, optimizing supply chains, and adapting to rapidly changing consumer demands. The study aims to investigate whether adopting closed-loop design principles, leveraging digital twin technology, and implementing e-libraries can provide innovative and effective solutions to these issues, thereby revolutionizing the traditional garment industry and shaping its future in a more sustainable, agile, and customer-centric manner.

Three-field plots and co-occurrence networks of author keywords play a vital role in bibliometric analysis. They provide valuable insights into the interconnections and patterns that characterize the academic literature and research.

The three-field plot analysis (Figure 1) consists of three components: the university affiliations of each co-author and the corresponding author (on the left), keywords associated with the *SCOPUS* database (in the middle), and the countries of affiliation for co-authors (on the right). This visualization provides a powerful means of representing the intricate relationships between these three variables. The height of the boxes within the three-field plot corresponds to the volume of publications generated by the respective affiliations, thereby offering a clear indicator of their research output. The taller the box, the greater the number of publications originating from that affiliation, which can be an important metric for assessing research productivity.

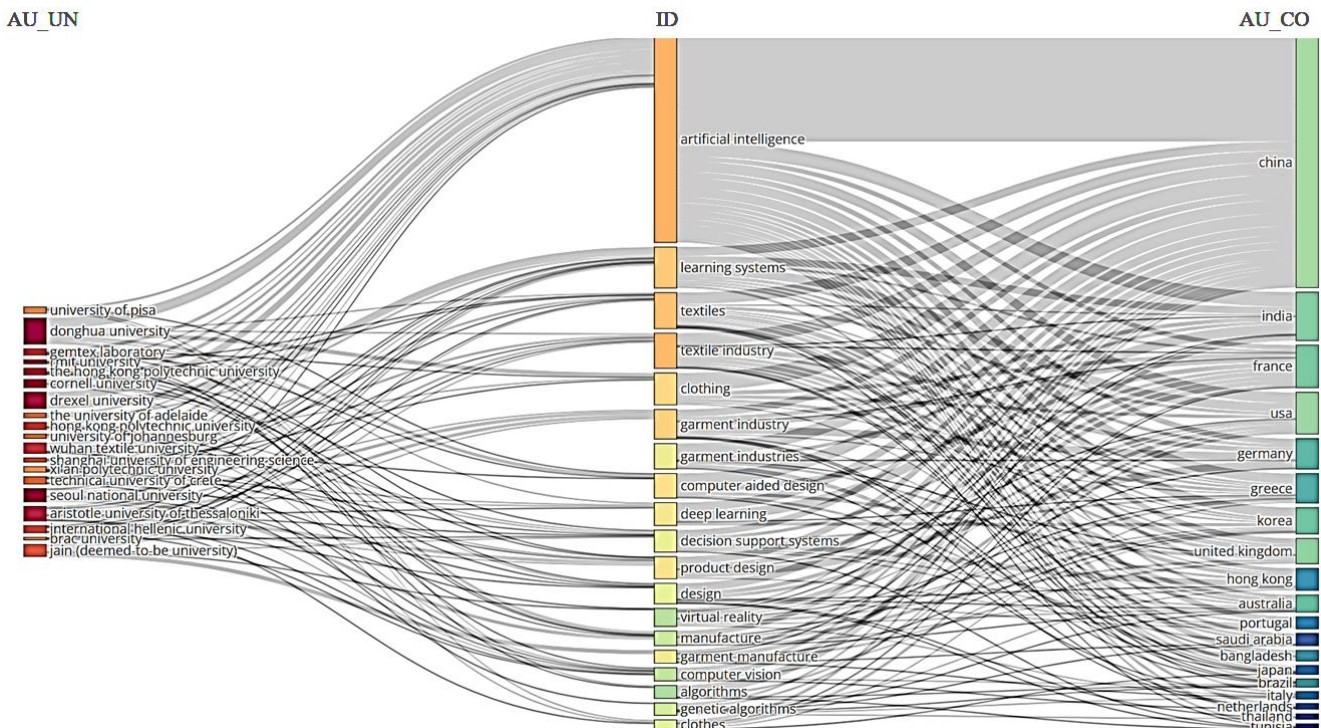

**Figure 1.** Three-field plot analysis (AU_UN—ID—AU_CO).

Co-occurrence network of author keywords is a graphical representation (Figure 2) that illustrates the relationships between author keywords in research publications. Within these networks, individual keywords serve as nodes, and the presence of connections (edges) between these nodes signifies the frequent co-occurrence of those keywords within the same research documents. These networks unveil valuable insights, offering a dynamic portrayal of the underlying thematic interplay within a specific research field or a distinct collection of publications.

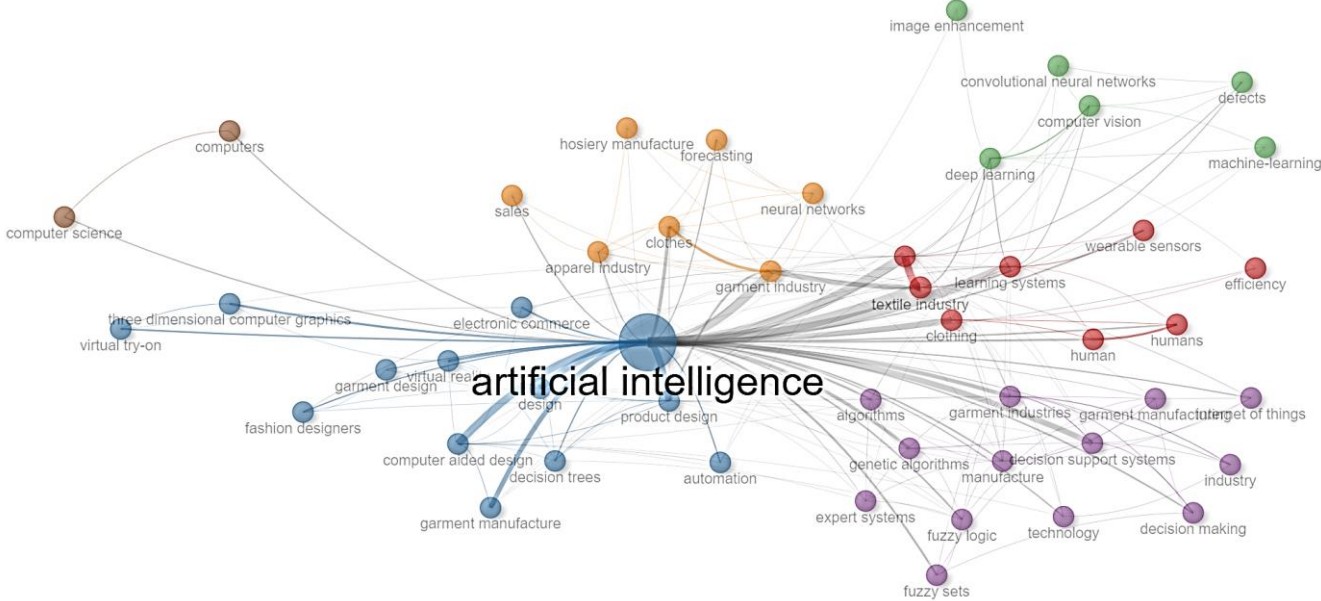

**Figure 2.** Co-occurrence network of author keywords.

Closer examination of Figure 1 reveals that, within the bibliometrics, China, the United States, France, and the United Kingdom prominently emerge as the forerunners, underscor-

ing the academic endeavors that chronicle the steps made within the artificial-intelligence-driven garment industry. Figure 2, specifically tailored to analyze keywords associated with artificial intelligence, offers a comprehensive analysis with a total of 50 nodes.

## 2. Advancements in 3D Body Scanning, Virtual Avatars, and Fashion Animation: A Transformative Fusion for Enhanced Virtual Experiences

This study explored the synergistic integration of 3D body scanning, virtual avatars, and fashion animation as a transformative combination that revolutionizes the fashion and virtual industries. Three-dimensional body-scanning devices, initially developed as non-contact body measurement solutions, were examined for their remarkable ability to capture intricate 3D geometry and human body architecture in fine detail, enabling integration into virtual environments. Diverse systems encompassing static or dynamic scan heads, single or multiple scan heads, handheld or turntable configurations, and portable or non-portable setups were explored, signifying the adaptability and versatility of this technology to cater to distinct application domains. Additionally, the emergence of body scanners with real-time motion capture capability, exemplified by the Move4D system, was investigated for its facilitation of efficient and realistic human avatar creation [3]. Emphasizing the implications for the fashion industry, the synthesis of these technologies propels innovation in garment design, personalized fitting experiences, virtual try-ons, and dynamic fashion presentations. Beyond fashion, this paper discusses the broader applications of this fusion in healthcare, entertainment, and virtual reality, elucidating the potential for personalized medical solutions, immersive experiences, and user engagement in various sectors. Ultimately, this paper contributes to the academic discourse surrounding the transformative possibilities brought forth by combining 3D body scanning, virtual avatars, and fashion animation and underscores their profound impact on enhancing virtual experiences across diverse industries.

## 3. Size World: Transforming Fashion Retail with Innovative Body-Scanning and Matching Technology

In the dynamic landscape of the modern fashion industry, the pursuit of the ideal fit has emerged as a dominant objective for both brick-and-mortar retailers and manufacturers alike. "Size World" has introduced a pioneering solution that integrates advanced technology, a refined matching algorithm, and an extensive digital framework, thereby piloting in a transformative shopping experience. The components and attributes encapsulated within "Size World" by utilizing state-of-the-art body-scanning technology encompass various facets that are meticulously designed to elevate the overall fashion retail landscape (Figure 3).

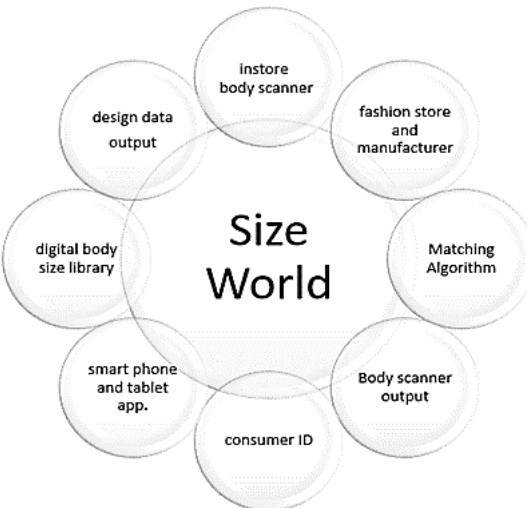

**Figure 3.** The Elements of "Size World" by State-of-the-Art Body-Scanning Technology.

Below is a breakdown of what each of these elements entails.

### 3.1. In-Store Body Scanner

This is a physical device located within a fashion retail store. It is equipped with technology such as sensors and cameras to capture the detailed body measurements of customers who use it. In-store body scanners provide a quick and accurate way to gather a customer's body dimensions, including height, weight, chest, waist, hips, and more. "Size World" incorporates state-of-the-art body scanning technology within fashion retail spaces and manufacturing facilities. These body scanners are designed to capture precise body measurements swiftly and accurately, providing a contactless experience for customers and efficient data collection for manufacturers.

### 3.2. Fashion Store and Manufacturer

In the context of "Size World," fashion stores and manufacturers play a fundamental role in implementing this technology. Fashion stores are the physical retail spaces where in-store body scanners are located. They offer customers the opportunity to use these scanners to obtain precise body measurements and personalized size recommendations. Manufacturers, on the other hand, collaborate with "Size World" to create clothing lines that align with individual customer needs. They rely on the matching algorithm, digital body-size library, and design data output to ensure a diverse range of sizing options, fabric choices, styles, and sizes that cater to a broad customer base.

### 3.3. Matching Algorithm

A matching algorithm is a computer program or set of rules designed to process and analyze data. In the context of fashion retail and body scanning, the matching algorithm takes the body measurements obtained from the in-store body scanner and compares them to a database of clothing sizes and styles. It uses this comparison to recommend the most suitable clothing sizes and styles for each customer based on their unique body measurements. At the heart of "Size World" lies a powerful matching algorithm. This algorithm takes the body-scanner output, including intricate body measurements, and pairs it with customer data, such as a unique consumer ID. The algorithm utilizes these data points to recognize the most appropriate clothing sizes and styles for each customer, guaranteeing an exceptional fit and delivering a personalized shopping experience.

### 3.4. Body-Scanner Output

The body-scanner output refers to the data and measurements collected by in-store body scanners when customers use them. It includes detailed information such as height, weight, chest, waist, hips, and more. The body-scanner output is the initial step in the process, providing the essential data that the matching algorithm and the digital body-size library use to generate personalized size recommendations. This output is the foundation of the "Size World" system, ensuring the accuracy and precision of the sizing recommendations.

### 3.5. Consumer ID

A consumer ID is a unique identifier associated with each customer who uses the "Size World" system. This ID helps in recognizing and distinguishing individual customers. It allows the system to link each customer's body measurements, preferences, and purchase history to provide a tailored shopping experience. The consumer ID ensures that the matching algorithm and the digital body-size library can offer accurate and personalized size recommendations, enhancing customer satisfaction and loyalty.

### 3.6. Smartphone and Tablet App

This is a mobile application that customers can download and install on their smartphones or tablets. The app serves as a bridge between the in-store body-scanning experience and the digital world. Customers can use the app to access and sync their body-scan data,

view size recommendations, explore the digital body-size library, and possibly even make purchases directly from their mobile devices. To bridge the gap between in-store experiences and the digital world, "Size World" offers a user-friendly smartphone and tablet app. Customers can easily download and access the app, which syncs with their body-scan data and personalized size recommendations. This app also allows shoppers to explore the digital body-size library and view design data for available clothing items.

### 3.7. Digital Body-Size Library

The digital body-size library is a comprehensive database that stores a wide range of body measurements and profiles. It includes data on various body types, sizes, and proportions. Retailers and manufacturers use this library to ensure that they have a diverse range of sizing options available to cater to a broad customer base. It helps in offering clothing that fits different body shapes and sizes. "Size World" maintains an extensive digital body-size library that encompasses a diverse range of body types and measurements. This library serves as a comprehensive reference, enabling retailers and manufacturers to cater to a wide customer base. It also ensures that every individual can find the perfect fit, fostering greater inclusivity within the fashion industry.

### 3.8. Design Data Output

This refers to the information and insights generated by the system based on the body measurements and preferences of customers. Manufacturers can use design data output to create clothing lines that are tailored to specific customer needs. This may involve adjusting patterns, fabrics, styles, and sizes to align with market demand and reduce production waste. For manufacturers, "Size World" offers design data output that is invaluable for creating clothing lines tailored to specific customer needs. By analyzing the data, manufacturers can adjust patterns, fabrics, and styles, ultimately reducing production waste and delivering garments that align with market demands.

Collectively, these components and features are part of an integrated system aimed at improving the shopping experience by providing personalized sizing recommendations, reducing the need for trial and error in finding the right fit, and enhancing the overall efficiency of the fashion retail and manufacturing process. "Size World" represents a model shift in fashion retail, combining innovation and data-driven precision to offer customers the perfect fit while enhancing operational efficiency for manufacturers. Through its integration of digital human modeling and virtual fitting rooms, "Size World" significantly reduces the need for physical prototypes. This reduction minimizes material waste, a fundamental step toward environmental sustainability. "Size World" facilitates the use of sustainable materials. This includes materials with lower resource consumption and environmentally friendly production processes, thus contributing to eco-conscious manufacturing. The technology enables AI-driven trend analysis, allowing manufacturers to anticipate market demands more accurately. This reduces overproduction, one of the fashion industry's notorious contributors to waste, and supports more ethical production practices. As "Size World" promotes individuality, it encourages consumers to make informed choices about their clothing, thus reducing the likelihood of impulse purchases. This conscious consumption aligns with sustainable fashion principles. By decreasing the need for physical shopping trips, "Size World" contributes to lowering carbon emissions associated with transportation and logistics. This, in turn, lessens the overall carbon footprint of the fashion industry.

Figure 4 provides a comprehensive overview of the crucial factors influencing an understanding of a customer's preferences and habits. It prominently features customer's historical body data, prior shopping behavior, the presence of a dress fit guide, and their level of engagement on various social media platforms. This combination underscores the diverse nature of comprehending a customer's preferences and habits, emphasizing the importance of a holistic approach.

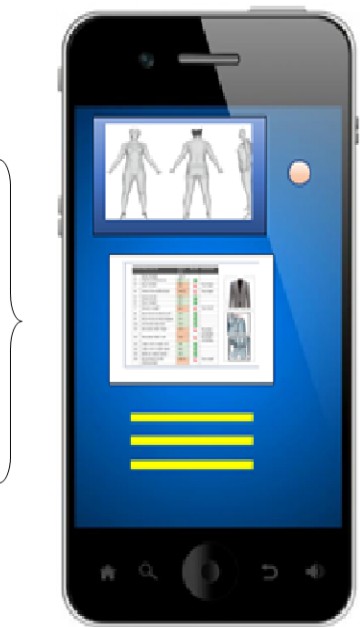

• Real-time information on whether the dress you buy is fitted to your body or not is provided.
• Important information such as the characteristics of the garment you are going to buy, the type of fabric, the type of production, and the origin is presented.
• Your body's recent-post data is made available.
• Your previous body information can also be viewed.
• Purchasing info records, including purchasing habits, are maintained.
• The combination of dress and online shopping, along with price information for the given period, planned delivery time, etc., is stored in the "Purchasing info record."
• The "Purchasing info record" is used to record the combination of dress and seller, along with price information for the given period.
• Health tips and recommendations based on your age and body shape are provided. • Advice or recommendations for diet, sports, or future situations are given.

**Figure 4.** Historical Body Data, Prior Shopping Behavior, Dress Fit Guide, and Social Media Engagement of Customers.

The historical body data highlight the significance of understanding a customer's physical measurements. Analyzing prior shopping behavior contributes to more informed marketing strategies. The presence of a dress fit guide offers a tailored shopping experience for customers. Finally, the customer's engagement on social media platforms underscores the importance of their online interactions and digital presence.

By integrating historical body data, shopping behavior analysis, dress fit guidance, and social media engagement tracking, businesses can refine their marketing strategies and product recommendations.

## 4. Establishing Uniformity in Body-Scanner Output and CAD Output

The drive for standardization in body-scanner output and computer-aided design (CAD) output holds significant implications across diverse industries, with fashion and manufacturing at the forefront. This effort is rooted in the fundamental goal of ensuring that data generated by body scanners adhere to a consistent format and contain uniform content. Such standardization plays a pivotal role in simplifying the interpretation and utilization of these data for design and production purposes.

One of the primary advantages of this standardization effort lies in its ability to facilitate communication among various participants within the fashion and manufacturing sectors. From manufacturers and designers to retailers and end customers, standardized data ensure that everyone speaks the same language when discussing body measurements and design specifications. This harmonization streamlines processes, minimizes errors, and fosters collaboration, ultimately leading to more efficient and effective operations.

Furthermore, standardization serves as the cornerstone for the accuracy and reliability of CAD systems. These systems heavily rely on precise body measurements to create customized clothing and prototypes. When the data from body scanners adhere to a standardized format, CAD systems can work with greater precision, resulting in clothing that fits perfectly and prototypes that closely match design specifications.

This solid commitment to standardization contributes to both efficiency and precision within the garment industry. This, in turn, leads to the production of higher-quality products that align closely with customer expectations. Consequently, the fashion and

manufacturing sectors witness not only enhanced productivity and reduced wastage but also a more satisfying and enjoyable customer experience.

As depicted in Figure 5, this initiative visualizes the establishment of uniformity in both body-scanner output and CAD output. It symbolizes a concerted effort to create a shared language and standardized framework within the industry, ultimately fostering innovation, quality, and customer satisfaction.

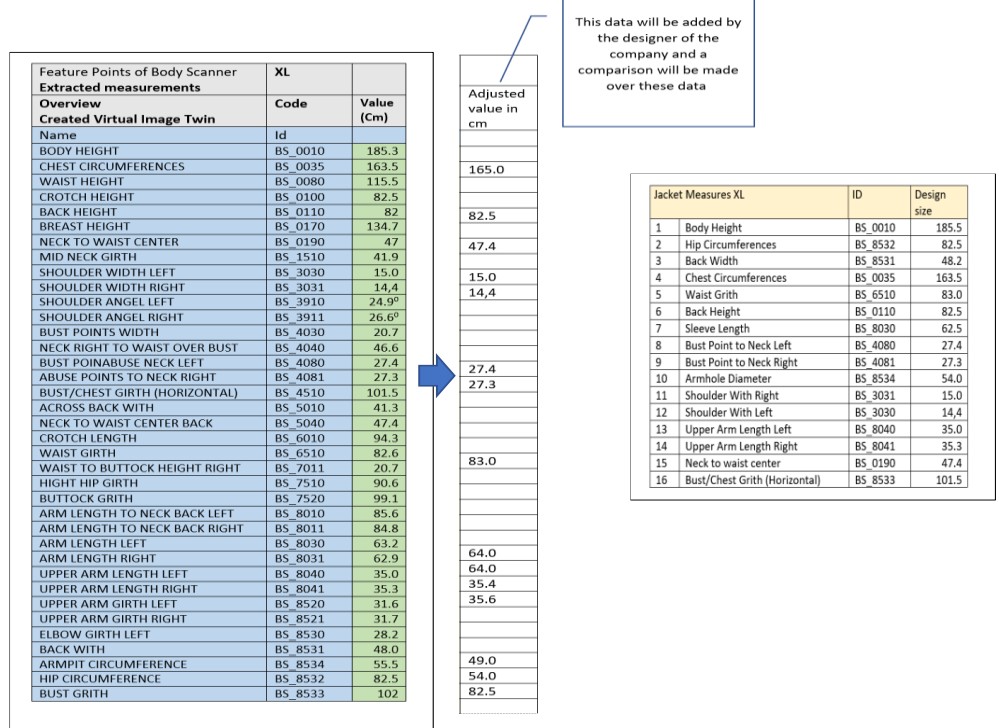

**Figure 5.** Standardization for Body-Scanner output and CAD output.

In order to streamline the design process and ensure alignment with the creative vision of the fashion firm, it is crucial to integrate the modified data provided by the firm's designers. These data act as a blueprint for creating garments that embody the designer's distinctive style and artistic direction. By incorporating these adjustments, designers and production teams can collaborate effectively, translating concepts into tangible fashion pieces that resonate with consumers. This collaborative approach not only enhances efficiency but also nurtures creativity, resulting in fashion collections that captivate and inspire. The blue colors in the scanner data, green colors on the scale (Figure 6a), and red colors represent comfort tolerances (Figure 6b).

During the digital mannequin design process, the fashion designer is responsible for specifying the adjustments required to align the dress model's measurements with those obtained from the body scanner. This integration involves incorporating plug-in data into the dress image, allowing for a direct comparison. Additionally, fit data should be meticulously added, accounting for factors such as body length and design type. For instance, the addition of 1 cm to each side of the shoulder width ensures a snug and precise fit. This meticulous attention to detail ensures that the final product harmoniously complements the wearer's unique body measurements and design preferences.

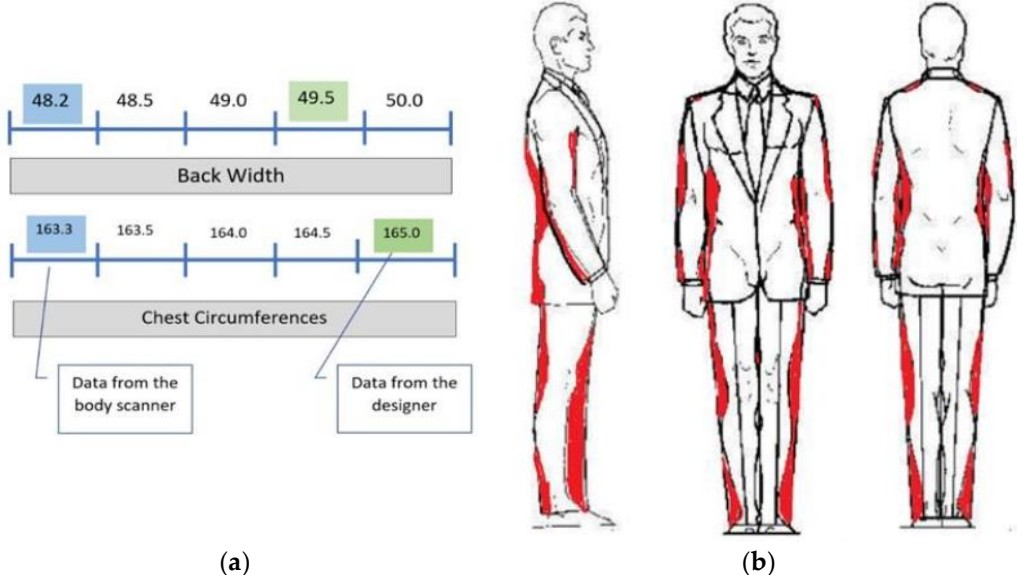

|  |  |
|---|---|
| (**a**) | (**b**) |

**Figure 6.** Integration of Data from Body Scanner and Designer (**a**) and Visualization of Comfort Tolerances in Red Color (**b**).

For each dress, the designer's tasks involve incorporating essential components:

Plug-In Data for Customer's Body-Size Chart: The designer should integrate plug-in data that correspond to the customer's body-size chart. This information serves as a foundation for ensuring that the dress fits the customer perfectly. The plug-in data encompass the unique measurements and proportions specific to each customer, enabling personalized garment recommendations and tailoring.

Plug-In Data for Dress CAD: In addition, the designer should include plug-in data or extension data related to the dress CAD (computer-aided design). These design data elements facilitate compatibility between the customer's chosen dress and the dress CAD data. Importantly, this integration does not necessitate altering the core host program, ensuring a smooth and efficient process for both the designer and the selected buyer.

By skillfully incorporating these plug-in data elements, the designer can create a customized and precise dress selection experience for each customer, optimizing both fit and design (Figure 7).

The world of fashion is continuously evolving, driven by a dynamic interplay of design aesthetics, technological innovation, and consumer preferences. Today, fashion designers are increasingly harnessing the power of digital tools and technologies to push the boundaries of creativity and craftsmanship. From 3D modeling and virtual fitting rooms to sustainable materials and AI-driven trend analysis, these advancements are revolutionizing the way fashion is conceived, produced, and experienced.

The successful integration of digital human modeling and virtual fitting rooms brings about a significant transformation in the production process by reducing the need for traditional physical prototypes. This reduction not only minimizes material waste but also enhances environmental sustainability. In the context of the fashion industry, this integration is particularly impactful, as it helps reduce the consumption of fabrics and other materials used in the development of textile products.

Furthermore, the utilization of sustainable materials and AI-driven trend analysis holds the potential to further reduce environmental impact in the fashion sector. The selection of sustainable materials typically involves production processes that consume fewer resources and are eco-friendly. Additionally, AI-supported trend analysis enables manufacturers to predict market demands more accurately and minimizes overproduction.

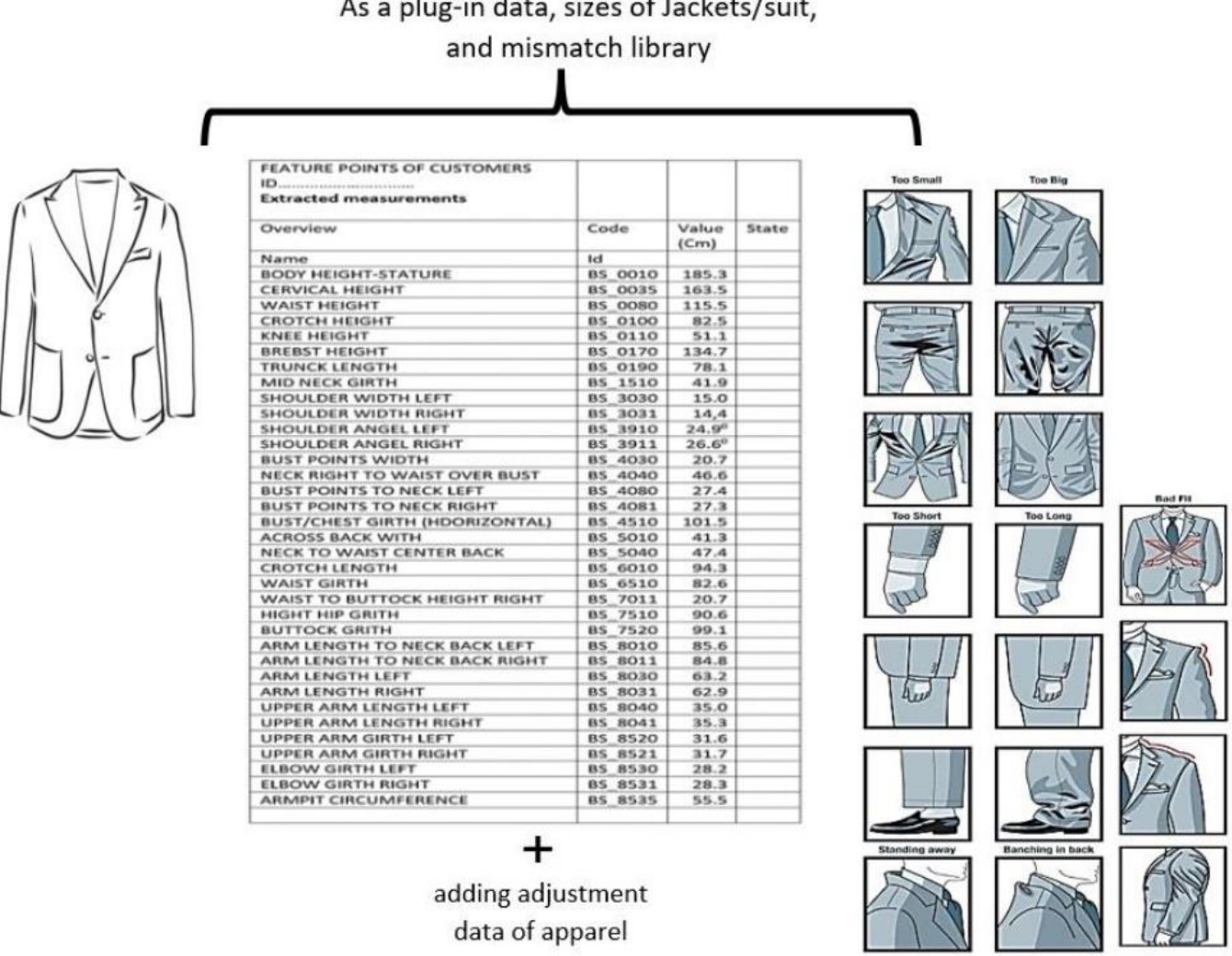

**Figure 7.** Jacket/Suit Sizes as Plug-In Data and Mismatch Library.

This synergy of art and technology promises to shape a fashion landscape that is more inclusive, responsive, and eco-conscious, catering to the diverse tastes and values of today's global consumers. The ultimate result, the form that will be presented to the customer is given in the following figure (Figure 8).

The comparison algorithm serves as a fundamental cornerstone in the realm of fashion technology. With its capabilities, it not only enables precise comparisons but also opens up the realm of personalized dressing within the context of online sales. This algorithm plays a pivotal role in deciphering the intricate balance between individual customer body measurements and the attributes of various clothing items. As a result, customers can look forward to a shopping experience that goes beyond generic recommendations, allowing them to curate their wardrobe with clothing items tailored specifically to their unique size, style, and preferences. This intersection of data-driven algorithms and the fashion industry is controlled to revolutionize the way we shop for clothing, fostering a sense of individuality and satisfaction among online shoppers. Figure 9 presents a visual representation of the comparison program, offering a comprehensive view of its functionality and outcomes.

| Jacket Measures XL | | Value (cm) | Results | Command |
|---|---|---|---|---|
| 1 | Body Height | 185.5 | | |
| 2 | Hip Circumferences | 82.5 | ✔ | |
| 3 | Back Width | 48.2 | ✘ | Too large |
| 4 | Chest Circumferences | 163.5 | ✘ | Too large |
| 5 | Waist Girth | 83.0 | ✔ | |
| 6 | Back Height | 82.5 | ✔ | |
| 7 | Sleeve Length | 62.5 | ✘ | Too short |
| 8 | Bust Point to Neck Left | 27.4 | ✔ | |
| 9 | Bust Point to Neck Right | 27.3 | ✔ | |
| 10 | Armhole Diameter | 54.0 | ✔ | |
| 11 | Right Shoulder Width | 15.0 | ✘ | Shoulder rumpling |
| 12 | Left Shoulder Width | 14,4 | ✘ | Shoulder rumpling |
| 13 | Upper Arm Length Left | 182 | ✔ | |
| 14 | Upper Arm Length Right | 94.3 | ✔ | |
| 15 | Neck to waist center | 47.4 | ✔ | |
| 16 | Bust/Chest Girth (Horizontal) | 101.5 | ✘ | Too small |

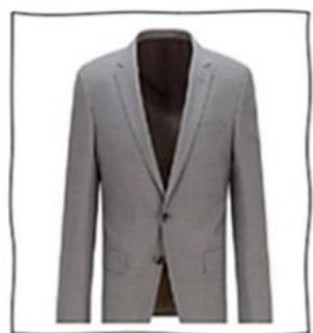
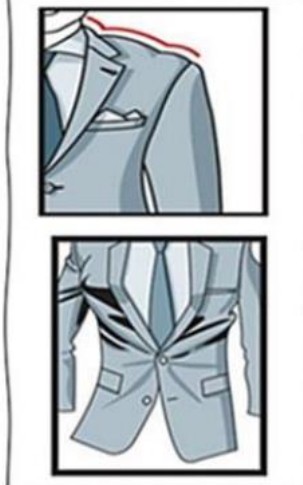

**Figure 8.** Visual Representation for the Customer.

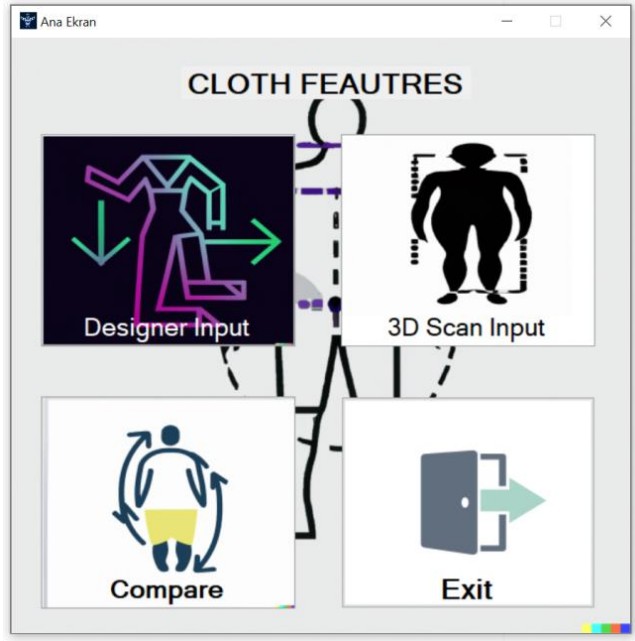

**Figure 9.** View of the comparison program.

This binary outcome of true or false, arising from changes to the data in the third row, holds significant implications within data analysis and decision-making processes (Figure 10). It serves as a critical indicator of the data's alignment with specific criteria or conditions set forth by the analytical framework. When the comparison yields a "true" result, it signifies that the altered data conform to the predefined criteria, suggesting coherence and compliance. Conversely, a "false" outcome signals a deviation from the established standards, highlighting discrepancies or incongruities within the dataset. This distinction aids analysts and decisionmakers in identifying and addressing data inconsistencies, ensuring the reliability and accuracy of subsequent analyses and conclusions. Such rigorous scrutiny and validation of data alterations are fundamental in maintaining the integrity of research and decision-support systems.

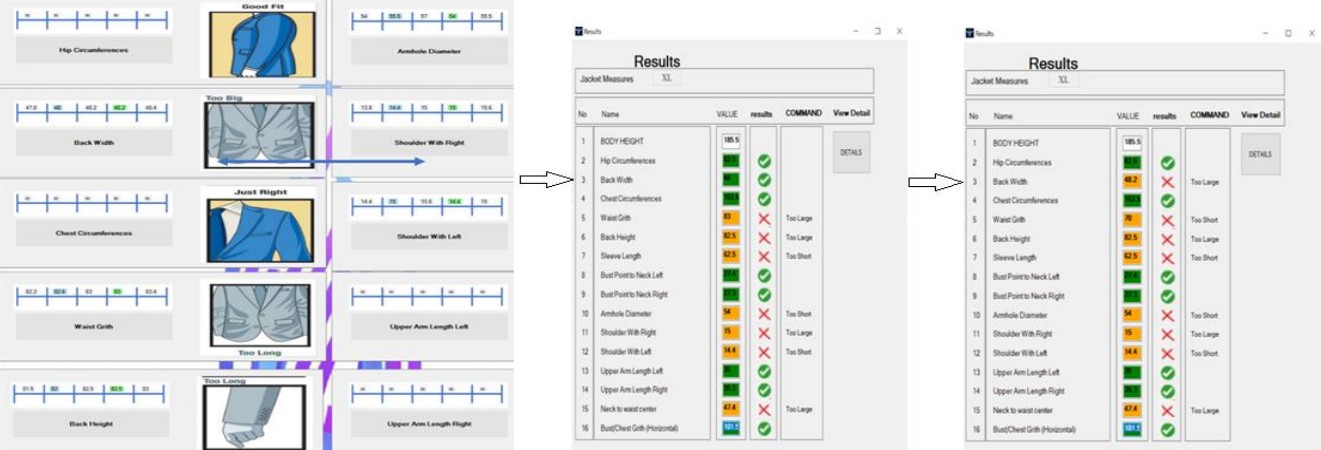

**Figure 10.** On-Screen Presentation of Results from Customized Programming.

In the context of Garment 5.0, the 3D body scanner serves as a crucial source of body data, marking a significant evolution in the fashion and apparel industry. This sophisticated technology not only captures precise body measurements but also contributes to several critical facets of garment development and customization. Firstly, it plays a central role in the realm of fit dress finding ergonomics, enabling designers and manufacturers to create clothing that aligns with the unique contours of an individual's body. Moreover, the 3D body scanner's capabilities extend to obesity determination, assisting in the assessment of body composition and size variations. Additionally, it aids in identifying and addressing body asymmetry, a crucial consideration for achieving optimal fit and comfort in garments.

Furthermore, the integration of the 3D body scanner extends to the mockup phase, where digital prototypes are created, refined, and validated. This process streamlines the development of garments by reducing the need for physical samples, thereby contributing to sustainability efforts in the industry. The scanner's data also populates the digital body library, serving as a valuable resource for designers and manufacturers. This library is a repository of diverse body shapes and sizes, facilitating the creation of garments that satisfy a broad spectrum of consumers. While standard sizing can serve as a useful starting point, offering customization options or personalized recommendations based on specific body measurements can further improve the fit and comfort of clothing.

As part of the fit dress application mockup stage, the 3D body scanner's data are utilized to create virtual mockups of clothing items. These digital representations allow for precise testing and adjustments to ensure a flawless fit and overall design. Ultimately, these advancements embody the lifecycle of customized digital systems in Garment 5.0, where technology not only enhances efficiency but also fosters greater inclusivity and sustainability in the fashion industry. Figure 11 illustrates the life cycle of a customized digital system within the context of Garment 5.0, including components such as the digital body library and the fit dress application mockup. This figure represents the various stages

and processes that the digital system, along with these components, undergoes from its initial creation and implementation to its eventual retirement or replacement.

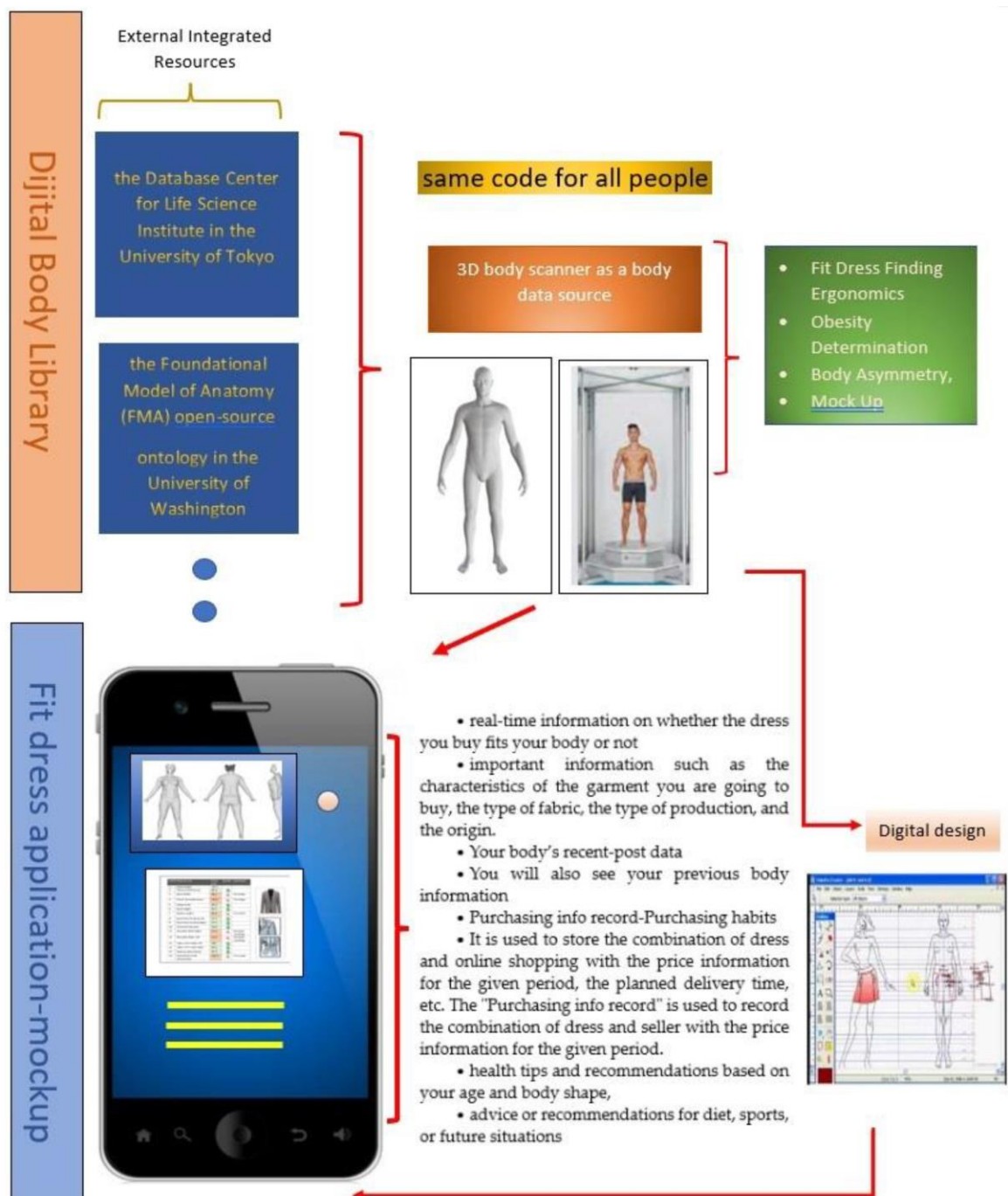

**Figure 11.** Life cycle of the customized digital system in Garment 5.0.

By analyzing data from the digital body library in real time, fashion companies can respond more rapidly to shifts in consumer preferences and emerging trends, ensuring they stay competitive in the market. The implementation of Q-codes serves as a versatile solution for both online and in-store shopping experiences. By utilizing this unique code, customers can transition between internet sales and physical retail locations. This integration eliminates the need for traditional fitting rooms, offering a more efficient and convenient approach to finding the perfect fit.

With their individual ID number codes, customers gain the ability to effortlessly determine which garments will suit their body measurements. This personalized approach not only saves time but also enhances the overall shopping experience. The Q-code system empowers customers to make informed choices, reducing the uncertainty associated with clothing fit and ensuring a more satisfying and confident purchase process. In this way, Q-codes bridge the gap between online and in-store shopping, offering a modern and customer-centric solution.

## 5. Case Study: Body-Size Avatar System with Customer ID

Customer ID 043/1-2-Sabina-24 represents a unique identifier within a system designed to enhance the customer experience in the fashion and clothing industry. This case study explored the output and implications of this specific customer ID.

Customer ID 043/1-2-Sabina-24 belongs to Sabina, a 24-year-old individual. The primary goal of the system associated with Customer ID 043/1-2-Sabina-24 is to provide a personalized and efficient shopping experience for Sabina and others with similar IDs. It aims to do so by utilizing historical body-size data, preferences, and shopping behaviors.

### 5.1. Personalized Clothing Recommendations

When Sabina logs into an online fashion store or visits a physical outlet, the system recognizes her as Customer ID 043/1-2-Sabina-24. This recognition allows the system to provide personalized clothing recommendations based on her past purchases, body measurements, and style preferences.

### 5.2. Efficient In-Store Experience

In the physical store, Sabina can simply provide her Customer ID, and store personnel can access her historical data to suggest clothing items that are likely to fit her well. This eliminates the need for time-consuming trips to the fitting room and enhances her shopping experience.

### 5.3. Online Shopping Ease

When shopping online, Sabina can enter her Customer ID, and the website or app will display a curated selection of clothing options tailored to her specific body size and style preferences. This streamlines her online shopping experience and increases the likelihood of finding items that meet her needs.

### 5.4. Size Accuracy

The system ensures that clothing items recommended or suggested to Sabina are chosen with precision, considering her body measurements recorded in millimeters (mm) or kilograms (kg). This significantly reduces the chances of ordering ill-fitting clothing items, reducing the need for returns and exchanges.

### 5.5. Customer Loyalty

Over time, as Sabina continues to use her Customer ID for shopping, the system accumulates more data about her preferences and sizes. This information helps the retailer build customer loyalty by consistently providing a high-quality and personalized shopping experience.

### 5.6. Privacy and Security

The system takes privacy and security seriously, ensuring that Sabina's personal and body-size data are protected. Access to these data is strictly controlled and only used for the purpose of enhancing her shopping experience.

Sabina experiences a convenient and personalized shopping experience both online and in store. The retailer reduces the rate of returns and exchanges due to improved size accuracy. Customer loyalty is fostered as Sabina consistently finds clothing that fits her well

and aligns with her style preferences. The retailer gains valuable insights into customer behaviors, helping them optimize their inventory and marketing strategies.

Customer ID 043/1-2-Sabina-24 highlights the capability of utilizing historical body-size data and preferences to shape a personalized and streamlined shopping experience. This strategy enhances convenience, ensures size accuracy, and fosters customer loyalty, all while placing a premium on privacy and security. It stands as a prime example of how technology can elevate and transform the fashion and clothing industry.

In a broader context, Customer Identity ID-X, which redefines personalized shopping experiences within the fashion industry, serves as an illustrative innovation. It provides tailored recommendations grounded in individual body measurements and preferences, thereby optimizing the shopping encounter, elevating customer loyalty, and guaranteeing sizing accuracy. This system not only enhances distinctive shopping experiences but also strikes a harmonious equilibrium by giving principal importance to privacy and security measures. This model underscores the transformative capacity of technology in the scope of fashion retail.

## 6. Body-Size Library: A Comprehensive Repository for Personalized Apparel Design and Virtual Avatars

During the body-measurement process using a 3D body scanner, approximately 40 body measurements are captured, with 20 focusing on the upper part of the body and 20 on the lower part. These precise measurements are considered sufficient by apparel manufacturers to create a well-fitted dress. The collected data are then stored as a standard representation of the individual's body, referred to as their "digital twin" avatar [3].

When a customer's body is measured, the resulting body-size information is first provided to the customer. Simultaneously, these data are also added to the digital library for future reference. Apparel manufacturers can access these data from the library to manage and tailor dress designs based on individual body proportions. It is essential for the avatar body size and the dress body size to match, ensuring that the data used by the manufacturer for dress design align with the data the customer uses for selecting the dress. This synchronization ensures a perfect fit, allowing both the manufacturer and the customer to determine whether the dress complements the individual's body shape effectively.

Additionally, the digital library offers a digital mockup virtual assembly test (VAT) for mechanical products. This feature aids in guiding virtual assembly testing before the physical production of the products. The VAT process allows manufacturers to assess the functionality and compatibility of mechanical components in a virtual environment, ensuring a streamlined and efficient production process [16].

The integration of 3D body-scanning technology, the digital twin avatar, and the digital library enables a data-driven approach to personalized apparel design. The precise body measurements collected from the 3D body scanner facilitate well-fitted dress designs, while the digital twin avatar serves as a standardized representation of the individual's body. The inclusion of a virtual assembly test further enhances the efficiency and accuracy of mechanical product development, contributing to successful manufacturing processes. The digital library was established with a commitment to safeguarding personal data in accordance with relevant data protection laws. It serves as a secure repository for various types of data, ensuring the privacy and confidentiality of its users. The following key aspects highlight the functionality and purpose of the digital library:

### 6.1. Protection of Personal Data

The digital library operates under stringent data protection measures to ensure the confidentiality and privacy of personal information. This includes the customer's body measurements, which are collected during 3D body scanning. The library strictly adheres to legal regulations, safeguarding the customer's sensitive data from unauthorized access or misuse.

### 6.2. Customer Access to Historical Data

Upon collecting the customer's body measurements, the library facilitates the transfer of this information to the customer's application. Through this process, the customer gains access to their own historical body-size data, which are presented digitally. This feature empowers customers to track and monitor changes in their body measurements over time, fostering a personalized and informed experience.

### 6.3. Utilization of Demographic Data

In addition to body-size measurements, the digital library incorporates demographic data, which include background information on users. These data encompass details such as sex, dwelling or working place, occupation, and education. By employing demographic data, the library can describe different members of the user population or specific population segments, enhancing the personalization and contextualization of services.

### 6.4. Anthropometry Study and Measurement

The digital library includes an anthropometry study that involves the comprehensive measurement of the physical dimensions and mass of the human body and its external parts. This study aids in creating detailed profiles of users' body sizes and shapes, forming the basis for accurate virtual representations in the form of avatars.

### 6.5. Anthropometric Data Collection

The library houses a collection of individual body measurements, known as anthropometric data, along with the accompanying background information recorded from a specific group of people, referred to as the sample. This database serves as a valuable resource for statistical analysis and research purposes, supporting the development of insights into human body-size variations and trends.

### 6.6. Anthropometric Report

As part of its comprehensive offerings, the digital library includes technical reports known as anthropometric reports. These documents provide a detailed account of the origin, contents, methodologies, and statistical characteristics of the anthropometric database. These reports are instrumental in ensuring transparency, reproducibility, and reliability in research and data analysis conducted within the library.

The sizing system for garments is primarily based on the International Organization for Standardization, with the EN 13402-4 [17] coding system commonly used to determine garment sizes, such as S-M-L-XL. This sizing system is typically recommended for sizing men's clothing, designed for five different body types. In contrast, women's clothing is typically sized to fit three distinct body types, often defined by measurements related to the hip, bust, and neckline. This approach aligns with ergonomic principles, where an ergonomic code system can be applied based on anthropometric data.

The digital library's (Figure 12) core principles revolve around the protection of personal data, customer access to historical measurements, and the utilization of demographic and anthropometric data for personalized services and research purposes. By upholding rigorous data protection measures and providing valuable insights into body-size data, the library serves as a crucial platform for advancing personalized apparel design and virtual avatar creation while respecting users' privacy and data security [18–22].

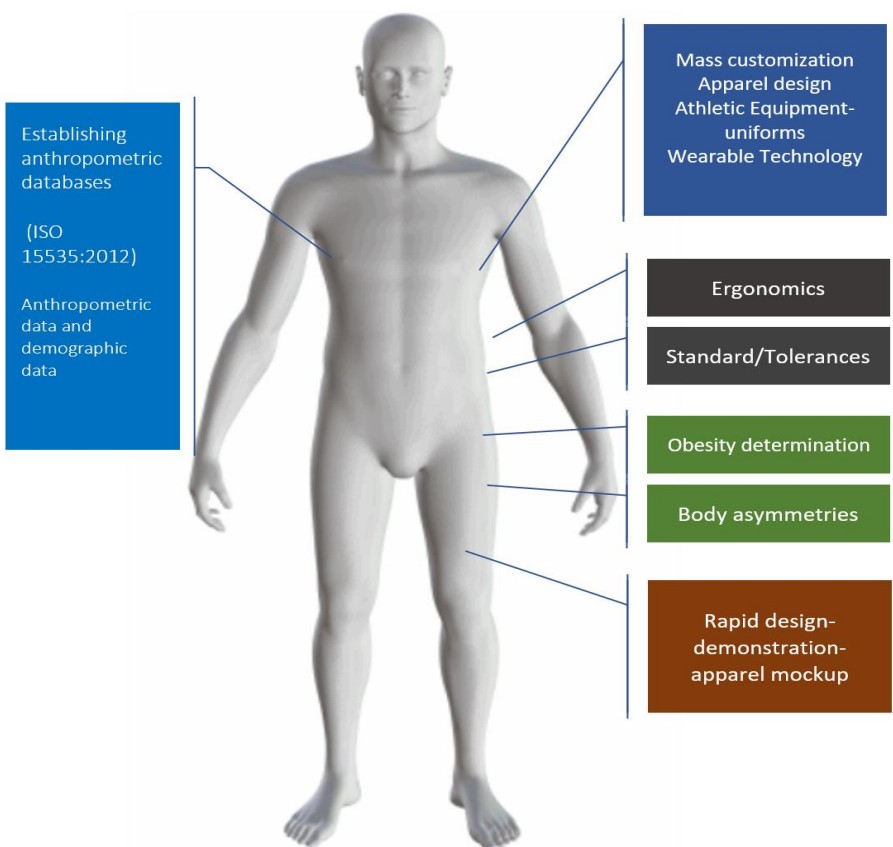

**Figure 12.** Digital Body-Size Library (DBSL) database content [19].

## 7. Integrating Robotics for Transformative Innovations into Garment Industry 5.0

For many years, automation has brought significant advantages to the industrial sector, ensuring consistent and precise workpieces while improving repeatability and accuracy in manufacturing. Economic justification has been primarily for large-scale production quantities, necessitating adaptive manipulation systems with robotics mechanisms. Robotics has expanded beyond its initial domain, finding universal application in both industrial and textile industries, yielding substantial benefits. Particularly in the textile sector, robotics is employed to reduce labor-intensive processes, and today, automation is synonymous with robotics, driven by computers enhancing efficiency and productivity, especially in textile-related applications [23].

The garment industry is witnessing a profound transformation with the advent of Industry 5.0, which emphasizes the integration of robotics and human expertise. This paper explores the promising prospects of integrating robotics to catalyze transformative innovations within Garment Industry 5.0 (GI5.0). Robotics in garment manufacturing, supply chain management, and retail operations is driving a revolution in the industry, enhancing efficiency, sustainability, and customer centricity.

Robotic applications in garment manufacturing are reshaping traditional production processes. Automated fabric cutting and sewing machines, collaborative robots (cobots) aiding skilled workers, and smart material handling systems are optimizing production lines, reducing lead times, and enhancing product quality. The fusion of robotics with digital technologies such as computer-aided design (CAD) and digital twins facilitates agile prototyping and customization, enabling on-demand and personalized garments [24].

Beyond the production floor, robotics is streamlining supply chain operations. Automated inventory management, intelligent warehousing, and robotic material handling improve logistics efficiency and accuracy. Real-time data analytics and robotic assistance in demand forecasting enhance supply chain visibility, reducing waste and inventory costs.

In the retail landscape, robotics are elevating customer experiences. Automated garment dispensers, smart fitting rooms with virtual try-on capabilities, and robot-assisted customer service optimize interactions, fostering personalized shopping journeys. Furthermore, robotic fulfillment centers enable efficient order processing and delivery in the era of omnichannel retail [25].

However, integrating robotics into GI5.0 poses challenges. Ensuring data security and privacy in the era of IoT-enabled robotics requires robust cybersecurity measures. Upskilling the workforce to operate, program, and collaborate with robots is essential to harness their full potential, ensuring a harmonious human–robotic collaboration. Embracing sustainability, the synergy between robotics and GI5.0 supports circular fashion principles. Robotics facilitate efficient recycling and repurposing of garments, minimizing waste and environmental impact.

The integration of big data in the apparel industry plays a pivotal role in providing personalized offerings to customers through e-commerce retailers. By analyzing large volumes of customer data, including preferences, purchasing behavior, and browsing history, apparel retailers can tailor their product recommendations and marketing strategies, enhancing the overall shopping experience for consumers. This data-driven approach allows for targeted promotions and personalized product suggestions, increasing customer satisfaction and loyalty. As a result, big data and AI combine forces to drive innovation and efficiency in the garment industry, paving the way for a more customer-centric and competitive marketplace [26].

The integration of robotics into GI5.0 signifies a significant shift in the fashion manufacturing landscape. For example, consider a garment production facility that incorporates robotic arms and automation for tasks like stitching, cutting, and quality control. These robots can work with unparalleled precision, ensuring that each stitch is consistent, and every piece of fabric is cut with minimal waste. Moreover, they can quickly adapt to changes in production demands. If the factory receives a rush order, the robots can seamlessly adjust their workflow to meet the deadline.

This level of agility and responsiveness is challenging to achieve with traditional manual labor. Additionally, the use of robotics reduces the need for extensive human intervention, minimizing the risk of errors and inconsistencies in the manufacturing process. As a result, the fashion industry is not only becoming more efficient but also moving toward sustainable practices. With reduced fabric waste and energy consumption and a lower carbon footprint, robotics in the garment industry paves the way for a more eco-friendly and sustainable future. Embracing the potential of robotics, this paper highlights the opportunities and challenges that will shape the future of garment manufacturing, retail, and supply chain management in the time of Industry 5.0 [27].

## 8. Integrating Closed-Loop Design for Transformative Innovations into Garment Industry 5.0

The adoption of closed-loop design principles is significantly associated with improved sustainability in the garment industry. A clothing manufacturer aims to improve the sustainability and efficiency of its production line by implementing a closed-loop design and digital twin system. They integrate digital twins, which are virtual replicas of the physical manufacturing process, to optimize the entire production cycle. The manufacturer uses the digital twin system to simulate different materials and assess their environmental impact. They make data-driven decisions to choose sustainable fabrics and materials, reducing the overall carbon footprint. Closed-loop design does not end with production. The digital twin continues to track clothing through distribution and inventory. The manufacturer can monitor inventory levels, assess customer demand, and implement just-in-time production strategies. This leads to fewer unsold items and reduced waste from overproduction.

The challenges faced by the apparel manufacturing industry are intensified by fast-changing fashion trends, increased product variety, and personalized demands. Quick and



optimized decision making is viewed as crucial to overcome these barriers. In the study, a methodology for the application of digital twin (DT) technology in apparel manufacturing plants was developed. The applicability and efficacy of the proposed method were demonstrated through a case study in which DT technology was implemented at a sewing assembly line. Real-time data were collected, and dynamic simulations were conducted to reduce bottleneck operations. The utilization of DT technology was shown to assist in decision making, allowing apparel manufacturing plants to respond swiftly to changing demands and reduce bottleneck operations. The effectiveness of the proposed methodology was demonstrated through the reduction of downtime and the improvement of production efficiency. Thus, it was concluded that the proposed method could enhance production efficiency, decrease downtime, and enable a quick response to changing demands within the apparel manufacturing industry [28].

In a period defined by technological innovation and sustainability imperatives, the garment industry finds itself at the link of transformation. Two powerful concepts have emerged to shape its future—"Closed-Loop Design using Digital Twin" and the flourishing "E-Libraries, Digital Twins, and Industry 5.0" revolution. These forces, individually remarkable, are now converging to forge a new path forward, one that holds immense promise for revolutionizing the way we create, produce, and consume clothing.

The concept of "Closed-Loop Design using Digital Twin" represents a groundbreaking approach to sustainability in the garment industry. By harnessing the capabilities of digital twins, manufacturers gain the ability to create virtual replicas of physical garments, enabling them to iterate and optimize designs with unprecedented precision. This approach reduces waste, minimizes environmental impact, and maximizes resource efficiency, all while maintaining a sharp focus on the end-user's needs and desires.

On the other hand, the advent of "E-Libraries, Digital Twins, and Industry 5.0" signals a new time for interrelation and data-driven decision making. E-libraries are reservoirs of knowledge, accessible at the fingertips of designers and industry stakeholders. They provide a rich tapestry of information, from historical fashion trends to sustainable materials, inspiring creativity while ensuring responsible choices. Meanwhile, GI5.0 takes automation and collaboration to the next level, fostering symbiotic relationships between human workers and intelligent machines.

What makes this convergence truly transformative is the potential for synergy between these two theories. As the garment industry aligns its creative processes with the principles of closed-loop design, digital twins not only facilitate sustainability but also serve as the bridge between the physical and digital spaces. These digital twins integrate with the vast knowledge repositories of e-libraries, enhancing design intelligence and innovation. Together, these elements hold the promise of a garment industry that is more agile, eco-conscious, and responsive to the evolving demands of a recognized global audience. Welcome to a future where technology, creativity, and sustainability unite to reshape fashion. Sustainability drives every choice, from materials to production. Fashion is no longer just about clothing; it is about responsible consumption, mindful production, and harmony with our planet.

## 9. Integrating Digital Twin for Transformative Innovations into Garment Industry 5.0

The integration of digital twins into the GI5.0 represents a profound shift in how clothing is designed, produced, and delivered. Digital twins, which are virtual replicas of physical garments and their associated processes, offer a dynamic and comprehensive approach to innovation within the industry. This academic exploration delves into the intricate facets of digital twins' transformative potential and their impact on GI5.0.

### 9.1. Design and Prototyping

Digital twins empower designers to create highly detailed virtual prototypes of garments. These digital replicas not only accelerate the design process but also enable designers to experiment with styles, sizes, and materials, fostering creativity and innovation.

### 9.2. Sustainability

Digital twins play a pivotal role in sustainability efforts. They facilitate the development of eco-friendly materials and production techniques by allowing for thorough simulation and optimization, reducing waste and energy consumption.

### 9.3. Supply Chain Management

Through real-time monitoring and analysis, digital twins enhance supply chain visibility and efficiency. Manufacturers can anticipate demand, optimize production schedules, and minimize disruptions, resulting in cost savings and improved resource allocation.

### 9.4. Personalization

Digital twins enable the creation of customized garments tailored to individual body measurements and style preferences. This level of personalization not only meets consumer expectations but also reduces returns and waste associated with ill-fitting clothing.

### 9.5. Quality Control

By simulating wear and tear on garments, digital twins aid in quality control. Manufacturers can identify potential issues before physical production, ensuring the delivery of high-quality products to customers.

### 9.6. Consumer Engagement

Virtual try-on experiences powered by digital twins enhance consumer engagement. Shoppers can visualize how clothing will fit and look on them, promoting online sales and reducing the need for physical store visits.

### 9.7. Challenges and Future Directions

While digital twins offer immense potential, challenges such as data security, cost, and integration into existing systems need to be addressed. Future research should focus on refining digital twin technology and expanding its applications.

The fashion industry faces challenges due to the lack of standardized sizing, leading to cross-brand inconsistencies that confuse online shoppers and result in unsuitable orders. Serial returns, where customers intentionally order more items than needed and return the rest, have a negative impact on retailers, causing some to raise prices to cover return costs. These returns often contribute to a significant carbon footprint as they pass through middlemen and resellers. Additionally, many returned items end up in landfills. Consumers value free shipping and returns, with 88% more likely to shop online when shipping is free and 68% considering free returns an incentive (Walker Sands' 2018 Future of Retail Report (https://www.walkersands.com/about/news/study-with-free-returns-consumers-twice-as-likely-to-spend-more-than-1000-online/, Last Access: 10 September 2023)). Unsatisfied customers are willing to pay more for tailored clothing due to sizing issues. Overall, the lack of standardized sizing and high return rates present challenges for the fashion industry, affecting customer satisfaction and environmental concerns.

In the context of implementing digital twins within the garment industry, a range of methodological approaches can be harnessed to address the intricate challenges tied to quantifying material savings, conducting economic cost–benefit analysis, and evaluating environmental metrics. These approaches are fundamental tools in the pursuit of sustainable and efficient garment production processes given in Table 2.

When focusing on quantifying material savings, several strategic pathways emerge. One pivotal approach involves real-time data integration, characterized by the deployment of IoT sensors and real-time data analytics. This approach allows for the continuous monitoring of material usage across the production process, affording precise measurement of both savings and waste. Additionally, using advanced data analytics and predictive modeling plays a significant role in optimizing material consumption. By analyzing historical data, patterns can be recognized, enabling informed decision making and ultimately cur-

tailing material usage. The introduction of material traceability through the integration of blockchain or RFID technology is equally crucial, providing transparent tracking of material origins and usage at each production stage. This bolsters accountability and transparency. Furthermore, investing in material efficiency software holds promise, offering actionable insights into the most effective use of materials to reduce waste and enhance resource management. The establishment of processes for recycling and reusing waste materials, including fabric remnants and off-cuts, is another sustainable practice that significantly diminishes the overall demand for materials.

**Table 2.** Methodological Approaches for Digital Twins in the Garment Industry.

| Aspect | Methodological Approaches | Digital Twin Tools |
|---|---|---|
| Quantifying material savings | — Real-time data integration (IoT sensors, data analytics)<br>— Advanced data analytics and predictive modeling<br>— Material traceability (blockchain or RFID technology)<br>— Material efficiency software<br>— Recycling and reusing waste materials | — Virtual sensor networks<br>— Material flow tracking<br>— Digital thread integration<br>— Three-dimensional fabric simulation<br>— Waste material tracking |
| Cost analysis | — Risk mitigation strategies<br>— Agility and market analysis<br>— Workforce development | — Financial modeling tools<br>— Risk analysis software<br>— Market trend analysis tools<br>— Workforce training software |
| Environmental metrics | — Emission tracking and reporting (including carbon footprint)<br>— Environmental impact assessment<br>— Circular economy implementation<br>— Compliance with regulations | — Environmental sensors<br>— Environmental impact analysis tools<br>— Circular economy optimization<br>— Compliance management systems |

Within the domain of economic cost–benefit analysis, a multifaceted approach is essential for the implementation of digital twins in the garment industry. Firstly, a comprehensive cost analysis is vital, encompassing not only the initial investment but also ongoing expenditures such as maintenance, software updates, and workforce training. In parallel, robust risk-mitigation strategies are required to identify and address potential risks. These strategies may encompass cybersecurity measures to safeguard sensitive data and the formulation of contingency plans to address operational disruptions. Equally critical is leveraging the agility enabled by digital twins. This involves adapting swiftly to evolving market dynamics through continuous monitoring of trends, consumer preferences, and global economic variables to refine production and distribution strategies. Complementary to these efforts is the allocation of resources for workforce development, empowering employees to effectively harness digital twin technology and cultivate a well-prepared and knowledgeable workforce capable of realizing the full potential of these innovations.

In the field of environmental metrics concerning the adoption of digital twins in the garment industry, several pivotal strategies can be identified. Emission tracking and reporting, anchored in real-time monitoring to trace and report greenhouse gas emissions, stands as a foundational approach. This approach capitalizes on data gathered from sensors and digital twin simulations, providing invaluable insights into the environmental impact of operations. Additionally, the comprehensive execution of a life cycle assessment (LCA) is indispensable. This assessment encompasses the holistic evaluation of the entire life cycle of garment production, with digital twins playing a central role in optimizing the environmental footprint at each stage. The implementation of circular economy practices is equally essential, promoting waste reduction and facilitating the recycling and remanufacturing of garments. Digital twins, with their responsiveness, significantly contribute to optimizing these processes for sustainability. Ensuring regulatory compliance is an imperative aspect, with digital twin technology supporting adherence to environmental regulations and standards, facilitated through the development of reporting

mechanisms that maintain compliance, mitigate legal issues, and uphold environmentally responsible practices.

The integration of digital twins heralds a transformative shift in Garment Industry 5.0 (GI5.0), with profound implications for design, sustainability, supply chain management, personalization, and consumer engagement. As technology continues to advance, the industry stands poised to harness the full transformative potential of digital twins, ushering in a new era of efficiency, creativity, and sustainability in fashion and garment manufacturing. In addition to the advancements facilitated by digital twins, one promising avenue for the garment industry may be the exploration of sustainable materials and production techniques. Embracing eco-friendly fabrics, such as those derived from recycled materials or sustainably sourced fibers, and adopting low-impact manufacturing processes could significantly reduce the industry's environmental footprint. This shift toward sustainable materials and practices not only aligns with the broader trend of eco-conscious consumerism but also presents an opportunity for the industry to distinguish itself and tap into the growing market for sustainable fashion. Such initiatives would not only enhance sustainability but also cater to the increasing demand for eco-friendly products.

## 10. Discussion and Conclusions

The hypothesis of this study revolves around the integration of digital human modeling (DHM), virtual modeling for fit sizing, ergonomic body-size data, and e-library resources into the garment industry. It suggests that this integration can lead to significant enhancements in the quality and personalization of clothing design and production. By using advanced technology and data-driven insights, the aim is to improve garment fit, increase consumer satisfaction, promote inclusivity, and streamline production processes. The expected result is a more efficient, environmentally responsible, and sustainable future for the garment industry, with the capability to provide tailored clothing options to consumers through a mobile application that interfaces with cloud-based data. The originality of this study is prominently highlighted by the development of a mobile application integrating with a cloud-based database. Its primary function is to offer users the most suitable garments based on their individual preferences, body size, and style choices. This innovative mobile app represents a significant departure from traditional garment shopping experiences, ushering in a new era of convenience and personalization for consumers.

The integration of e-libraries, digital twins, and GI5.0 has ignited a transformative revolution within the garment industry. These groundbreaking technologies have granted manufacturers the ability to craft tailored and perfectly fitted garments, capitalizing on data-driven strategies to cater to the ever-evolving preferences of consumers. The industry's renewed emphasis on efficiency, personalization, and sustainability positions it to flourish within this new landscape of technological advancements, promising a future of innovation and progress in the realms of fashion and apparel.

The beneficiaries of the research study encompass a diverse range of shareholders within the fashion and textile industry. This includes fashion designers and innovators seeking creative and efficient design tools, apparel manufacturers aiming to enhance production efficiency, sustainability advocates focused on reducing the industry's environmental footprint, supply chain managers striving to optimize global supply chains, retailers interested in improving inventory management and customer experiences, technology providers developing innovative solutions, consumers benefiting from improved product quality and sustainability, and academics and researchers contributing to the understanding of technology adoption and innovation in the fashion sector. Exploring the distinct needs and priorities of these beneficiaries is crucial in evaluating the impact of closed-loop design, digital twins, and e-libraries on the garment industry.

This makes possible a world where fashion is a force for good, where garments are not just beautiful but also meaningful, and where consumers can feel proud of the choices they

make. Welcome to a future where fashion reflects our values, as a celebration of innovation and a commitment to a sustainable and bright tomorrow.

The results suggest that essential studies provide significant advantages to users of the garment app through the integration of digital human modeling (DHM), virtual modeling for fit sizing, ergonomic body-size data, and e-library resources. The combination of DHM, virtual modeling, ergonomic body-size data, and garment library resources for individuals within a specified timeframe holds substantial promise for the future, especially if estimators for garment suitability for individuals are developed. This underscores the potential for integrated technologies to enhance personalized garment recommendations and fit assessments.

In terms of future implications, this study suggests that body e-libraries have the potential to serve as reliable health indicators for individuals. The comprehension of anatomical regions [29–31] remains fundamental for elucidating the intricate structure and functionality of the human body. Such knowledge continues to play a pivotal role in facilitating medical professionals and researchers in the accurate diagnosis and effective treatment of a wide range of health conditions and diseases.

With respect to future suggestions, within this innovative concept, every shopping center will implement a system where individuals undergo thorough body scanning to create a digital twin. This scanning process is designed to capture precise measurements and details of an individual's physique, creating a virtual representation. Subsequently, these individuals will gain access to a unique and personalized shopping experience facilitated through the cloud-based infrastructure. The cloud will serve as a central repository for a wide array of custom-made garments and apparel options, each tailored to suit the specific body measurements and preferences of the customers' digital twins. The scanned body data, containing vital measurements and personal style preferences, will be securely stored in the cloud and linked to the respective digital twins. To ensure accuracy and relevance, these stored digital twins will be subject to periodic updates at pre-defined intervals, thereby accommodating any potential changes in an individual's physique or preferences over time. This dynamic approach to clothing and fashion retail offers a tailored shopping experience, reflecting the evolving needs and choices of individuals and their digital twins.

**Author Contributions:** Conceptualization, S.D., P.D., I.B., G.B. and M.N.D.; methodology, S.D., P.D., I.B., G.B. and M.N.D.; investigation, S.D., P.D., I.B., G.B. and M.N.D.; resources, S.D., P.D., I.B., G.B. and M.N.D.; writing—original draft preparation, S.D., P.D., I.B., G.B. and M.N.D.; writing—review and editing, S.D., P.D., I.B., G.B. and M.N.D.; visualization, S.D., P.D., I.B., G.B. and M.N.D. All authors have read and agreed to the published version of the manuscript.

**Funding:** This research received no external funding.

**Data Availability Statement:** Please contact the authors for your data and model requests.

**Conflicts of Interest:** The authors declare no conflict of interest.

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
