# Peer review of "Revolutionizing the Garment Industry 5.0: Embracing Closed-Loop Design, E-Libraries, and Digital Twins"

_sustainability, doi:10.3390/su152215839_

Round 1

Reviewer 1 Report

Comments and Suggestions for Authors

I am grateful for the possibility to become familiar with a manuscript, which has the potential to become an excellent publication. The topic is very timely, and the particular subject-matter of the article, as well as the analysis are novel and very informative. My only remark is to improve the abstract.

The current abstract of the article does not quite correspond to the IMRAD scheme. A good, catchy but thorough abstract that briefly, concisely and clearly summarizes the main objective, basic goals, method(s), results and conclusions of the paper is a sine-qua-non condition. Its structure should follow closer the above  scheme and highlight major findings. The authors should note to what extent it would be beneficial: when searching in a database, the abstract is a basis for the main decision whether to download the paper, read it and—eventually—quote it.

Author Response

Dear Reviewer, I appreciate your positive feedback on the manuscript and your insightful comments regarding the abstract. Your suggestion to align the abstract more closely with the IMRAD scheme and to emphasize the main objectives, methods, results, and conclusions is well-taken. A well-crafted abstract is indeed crucial for effectively conveying the essence of the research to potential readers and researchers searching in databases. I will ensure that the abstract is thoroughly revised to better encapsulate the core elements of the study. In the manuscript, I made additions/corrections in yellow. Thank you for your valuable input, which will contribute to the overall quality of the paper. Best regards.

Reviewer 2 Report

Comments and Suggestions for Authors

This is not a research paper but a description of a project. The format has to be completely changed - none of the sections presented in the paper comply with their original purpose in a research paper.  

The project is very good and the description is interesting in terms of information, however the contribution to scholarship is unclear in the present format 

Given the journal’s scope, the authors should place the whole discussion under the notion of sustainable development. Thus, a short discussion of the term should be provided in the introduction. In this vein, the following two papers should be included. (a) Manioudis, M. & Meramveliotakis, G. (2022) “Broad strokes towards a grand theory in the analysis of sustainable development: a return to the classical political economy”, New Political Economy, 27(5), pp. 866-878, and (b) Tomislav, K. (2018) “The concept of sustainable development: From its beginning to the contemporary issues”, Zagreb International Review of Economics & Business, 21(1), 67-94.

One of the two main shortcomings of the reviewed manuscript concerns the theoretical background of it. This component of the study should be significantly extended with recent studies from the international literature, reflecting a logic and arguments with a critical approach.

Comments on the Quality of English Language

Moderate editing of English language required

Author Response

Dear Reviewer, I want to express my gratitude for your thoughtful feedback and constructive suggestions. We have thoroughly reviewed our entire paper, taking into account all the deficiencies you pointed out, and highlighted the corrections in yellow. Best regards.

Reviewer 3 Report

Comments and Suggestions for Authors

The authors present a quite interesting article about modern technologies that affect or change the garment industry. However, the main problem of the article for the journal Sustainability is its descriptive character. Chapter 1 to chapter 4 are only descriptive. Although it is a useful and summarizing description, I do not think that this description should be published in Sustainability, which is 1) a scientific journal that should solve, discuss, investigate scientific problems, questions and issues, 2) the journal primarily orientated on the issues of sustainability. Although the word “sustainability” or similar words are several times mentioned in the text, the analysis is quite general and its mine findings can be summarized as “new technologies contribute to the sustainability of garment industry”. For instance:
– table 1 containing a quite general description,
- sentence on rows 143-147: “With its body scanning technology, matching algorithm, and comprehensive digital infrastructure, "Size World" is poised to redefine how we experience and shop for fashion in both physical stores and online spaces. It's a solution that promotes individuality, sustainability, and customer satisfaction, shaping the future of fashion retail and manufacturing”,
- sentence on rows 238-242 “From 3D modeling and virtual fitting rooms to sustainable materials and AI-driven trend analysis, these advancements are revolutionizing the way fashion is conceived, produced, and experienced. As fashion becomes more democratized and personalized, consumers can expect a future where clothing is not just a statement of style but also a reflection of individuality and sustainability”. How exactly sustainability changes, increases, etc. is missing.

Chapter 5 contains “case study”, but it again contains a summary of the advantages that technologies offer to a customer. Yes, it is an interesting description and the possibilities of customers really change due to the described technologies, but should this be the topic of a scientific article? For a popular journal, it can be a very interesting topic, but I have serious doubts if the topic is suitable for a scientific one.

Sixth chapter concentrates on Body-Size Library, but it again only describes its functions, possibilities and aspects. Aspects of sustainability are not analyzed in this chapter at all – some aspects are mentioned in Table 1 but without deeper investigation.

Chapter 7 deals with the involvement of robotics in garment industry. I appreciate here that some challenges that the integration brings are mentioned, but again the chapter is too general, and descriptive without specific data. It is also true for the issue of sustainability – see rows 483-486 “The integration of robotics into GI5.0 marks a transformative shift towards data-driven, agile, and customer-centric fashion manufacturing. Through the precision, efficiency, and adaptability of robotics, the garment industry is on the brink of groundbreaking innovation and a new era of sustainability.”

Chapter 8 concentrating on Closed-Loop Design is unclear. The rows 493-503 probably describe the study mentioned in Reviews as 12, but it is hardly understandable what the study is about and to find any specific data. See “In the study, a methodology for the application of Digital Twin (DT) technology in apparel manufacturing plants was developed. The applicability and efficacy of the proposed method were demonstrated through a case study in which DT technology was implemented at a sewing assembly line. Real-time data were collected, and dynamic simulations were conducted to reduce bottleneck operations. The utilization of DT technology was shown to assist in decision-making, allowing apparel manufacturing plants to respond swiftly to changing demands and reduce bottleneck operations. The effectiveness of the proposed methodology was demonstrated through the reduction of downtime and the improvement of production efficiency. Thus, it was concluded that the proposed method could enhance production efficiency, decrease downtime, and enable a quick response to changing demands within the apparel manufacturing industry [12].” What do these sentences exactly tell?

The rest of the chapter is closest to the issue of sustainability, however only on a general level. If the article specified, how exactly digital twins contribute to waste reduction, it would be seen as a scientific finding. However, the article stays at too general level. A similar conclusion can be written about chapter 9 which only summarizes the possibilities of digital twins – again only on a general level, without any numbers and specific details.

Generally, although the article gives an interesting summarization of new technological trends in garment industry, I think that its scientific value is low, and its connections with the issue of sustainability are insufficient. The article can be, in its present form, published in a popular journal, but I do not recommend its publishing in the journal Sustainability.

Comments on the Quality of English Language

I have no major commens on the quakity of English language.

Author Response

(The authors gave the same response as above.)

Reviewer 4 Report

Comments and Suggestions for Authors

This paper aims to study the fusion of digital human modeling (DHM), virtual modeling for fit sizing, ergonomic body size data, and e-library resources in order to increase the quality of garment industry.

This research is based on the results obtained in the PhD thesis by one of the authors.

I would like to make the following recommendations:

- The abstract should present the aim, the objectives and the hypothesis of the research,; the methodology; the main results. 

- The authors should place their research wirhin the context of current and previous studies. They should include their research in the literature review background. 

- The authors should clearly present their research hypotheses. 

- In the Conclusion section, the authors should clearly underline the originality of their study. 

- It should be highlighted who are the beneficiaries of the findings from thid study (the customers, the manufacturers?). What is the profile of these beneficiaries? 

- The authors should present the ways of adopting their  research results by manufacturers. 

- Data Availability Statement is not completed. 

Author Response

(The authors gave the same response as above.)

Reviewer 5 Report

Comments and Suggestions for Authors

It is a narrative or descriptive Paper. 

Abstract should be revised to present the objective and Maine results.

What is the problem to address?

Figures, tables and développement, are not referenced at all.

Literature review is poor, to not say not relevant. It seems like a very good descriptive paper for an operation with no proved scientific value.

Should be revised et enriched with relevant literature review and clear methodology.

Comments on the Quality of English Language

Well writen 

Author Response

(The authors gave the same response as above.)

Round 2

Reviewer 2 Report

Comments and Suggestions for Authors

The authors addressed all my comments. 

Author Response

Dear Reviewer,

I wanted to express our sincere gratitude for your insightful comments and feedback on our work. Your guidance and suggestions have significantly contributed to the improvement of our manuscript. We truly appreciate the time and effort you dedicated to reviewing our work and providing us with valuable insights. Once again, thank you for your invaluable contribution to our research.

Best regards.

Reviewer 3 Report

Comments and Suggestions for Authors

The authors made many changes concerning to the article. However, my main objection mentioned in the first round of review was not adequately solved – the article still stays too general and too descriptive. There is no exact data describing how a specific technology (for instance digital twin) contributes to sustainability, for instance how many materials were saved, how it affected the costs of companies operating in garment industry and so on. From that perspective, my opinion that the article should be published in some more popular or less scientific journal does not change. For publication in the journal Sustainability, I recommend at least making some specific estimation (numbers) showing the impact of the described changes on Sustainability.

I did not find any additional value concerning the new part between rows 66 and 104 including Figures 1 and 2. If other reviewers do not request this information, I recommend deleting this part.

Comments on the Quality of English Language

No objections.

Author Response

Dear Reviewer,

I would like to express our heartfelt gratitude for your invaluable feedback and the constructive insights you have generously shared. Your meticulous explanations and thoughtful assessments have greatly enhanced the clarity and coherence of our paper. Regrettably, we are unable to incorporate data pertaining to Digital Twins at this time, as we are currently in collaboration with multiple companies and are on the verge of initiating Digital Twin applications. Nevertheless, we have diligently integrated the suggested corrections, which are noticeable in red within the manuscript.

Yours sincerely.

Reviewer 5 Report

Comments and Suggestions for Authors

Thank you for the update. Still have some points:

1-Figure 3: the indicate source. You according to Size World, but no specific document in list of references to precise the source.

2-Figure 3 has 8 cases, you explain only five, why?

3- Since you are considering a case study, one person only does not make a valid case study to be able to generalize the finding. Why not to consider un reel sample from 15 to 20 persons et consider other variables: age, weight, gender, etc....

Conclusion still long. can be divided between: Discussions and conclusion.

Comments on the Quality of English Language

Need some work.

Author Response

Dear Reviewer,

I wish to express our sincere gratitude for your thoughtful feedback and constructive suggestions. We have diligently reviewed our entire paper, addressing all the deficiencies you pointed out, and have highlighted the corrections in red for your reference.

Best regards.

Round 3

Reviewer 3 Report

Comments and Suggestions for Authors

I  am not still sure whether the article should be published in the journal Sustainability. It is still from my point of view too general, although it contains a relevant survey of new technologies in garment industry including problems that are connected with their introduction. I appreciate, here the last revision that includes some essential issues as the behavior of customers ordering many items, some of them just for trial and then returning them. But the article still should be more specific there. However generally I do not want to block publishing the article, so my final decision is “accept”.

The Reviewer